# Influence of Socioeconomic Status on SARS-CoV-2 Infection in Spanish Pregnant Women. The MOACC-19 Cohort

**DOI:** 10.3390/ijerph18105133

**Published:** 2021-05-12

**Authors:** Javier Llorca, Carolina Lechosa-Muñiz, Lorena Lasarte-Oria, Rocío Cuesta-González, Marcos López-Hoyos, Pilar Gortázar, Inés Gómez-Acebo, Trinidad Dierssen-Sotos, María J. Cabero-Pérez

**Affiliations:** 1Faculty of Medicine, Universidad de Cantabria, 39011 Santander, Spain; carolina.lechosa@scsalud.es (C.L.-M.); marcos.lopez@scsalud.es (M.L.-H.); pgortazar@gmail.com (P.G.); ines.gomez@unican.es (I.G.-A.); trinidad.dierssen@unican.es (T.D.-S.); mariajesuscabero@gmail.com (M.J.C.-P.); 2CIBER Epidemiología y Salud Pública (CIBERESP), 28029 Madrid, Spain; 3Hospital Universitario Marqués de Valdecilla, 39008 Santander, Spain; lorena.lasarte@scsalud.es (L.L.-O.); rocio.cuesta@scsalud.es (R.C.-G.); 4IDIVAL Instituto de Investigación Sanitaria Valdecilla, 39011 Santander, Spain

**Keywords:** COVID-19, SARS-CoV-2, pregnancy, household transmission, socio-economic status

## Abstract

Little is known on socio-economic factors associated with SARS-CoV-2 infection in pregnant women. Here, we analyze the relationship between educational, occupational, and housing variables with SARS-CoV-2 infection in a cohort of 988 pregnant women in Spain. Pregnant women were recruited at the University Hospital Marques de Valdecilla, Santander, Spain, among those delivering from 23 March 2020 onwards or consulting for their 12th week of pregnancy from 26 May 2020 onwards. Information on occupational variables and housing characteristics was self-reported. Pregnant women were tested for a current or past infection of SARS-CoV-2 using both PCR and antibodies detection (ELISA). Logistic regression models were used to analyze factors associated with SARS-CoV-2 infection, adjusting for age and country of origin. Infection by SARS-CoV-2 was not associated with educational level or occupational variables, except for where the pregnant woman was a healthcare worker (odds ratio (OR) = 2.87, 95% confidence interval (CI): 0.84–9.79). Housing with four or more rooms (OR = 2.07, 95% CI: 0.96–4.47), four or more people in the household (OR = 1.91, 95% CI: 0.89–4.14), lack of heating (OR = 2.81, 95% CI: 1.24–6.34) and less than 23 square meters per person (OR = 3.97, 95% CI: 1.43–11.1) were the housing characteristics associated with SARS-CoV-2 infection. Housing characteristics, but not occupational or educational variables, were associated with SARS-CoV-2 infection. Guidelines on the prevention of COVID-19 should reinforce household measures to prevent pregnant women from becoming infected by their relatives.

## 1. Introduction

Several studies have reported a socio-economic gradient in SARS-CoV-2 infection, hospitalization and death. Lower educational level, living in a rented accommodation and households with lower average income were associated with being more likely to require testing [1]. Risk of infection was higher in the most deprived and unskilled people [2], those living with more people, renting or owning with a mortgage or with lower educational attainment [1]. Being admitted to hospital because of COVID-19 was more frequent in people of lower socio-economic status [1], and mortality was higher in the most deprived [3].

Pregnancy put women infected with SARS-CoV-2 in danger of severe disease, especially if infected in the third trimester [4], leading to more frequent admission into ICU and higher rates of stillbirth than uninfected pregnant women [5]. However, a “paucity” in studies on pregnant women has been highlighted [6]. Little is known about socio-economic status and SARS-CoV-2 infection in pregnant women. An ecological study communicated that pregnant women living in neighborhoods with higher unemployment rates, large household membership and lower median incomes had a higher risk of getting infected [7], although it had a small sample size (n = 434), and a high degree of correlation between several neighborhood-level variables disallowed multivariate analysis.

In this article, we study the association between socio-economic status—as measured by educational level, employment and housing characteristics—and infection by SARS-CoV-2 during pregnancy in a cohort of Spanish women.

## 2. Methods

### 2.1. Setting and Patients

The Mother And Child COVID-19 (MOACC-19) cohort is described elsewhere [8]. In brief, we recruited all pregnant women attending the University Hospital Marques de Valdecilla, Santander, Spain. This hospital treats patients not only from the city of Santander (about 170,000 inhabitants) but also from an area of influence that includes several smaller urban nuclei as well as many rural areas (about 320,000 inhabitants). Moreover, in the first months of the pandemic, most pregnant women in the whole region (about 600,000 inhabitants) were treated at this hospital. Recruitment began on 26 May 2020. Women were invited to participate in three ways: (1) Retrospective sub-cohort: Women admitted for delivery from 23 March 2020 to 25 May 2020 were invited by phone. They all had a PCR test performed on the day of their admission. (2) Prospective sub-cohort: Women admitted for delivery from 26 May 2020 onwards were invited at admission. (3) Prospective sub-cohort of ongoing pregnancies: Women attending their standard consultation at the 12th week of gestational age from 26 May 2020 onwards were invited at that consultation.

### 2.2. Ethical Issues

The study was approved by the Committee on Ethics in Clinical Research of Cantabria, Spain (certificate no. 2020.174). All participants signed an informed consent form.

### 2.3. Information and Variable Managing

Information on reproductive history (number of deliveries, history of abortions), current pregnancy (way of fertilization, gestational age, education on pregnancy, toxic habits, vaccines, gestational pathology) was obtained from medical records.

### 2.4. Main Exposure: Socio-Economic Status

Data on socio-economic status were obtained by interviewing the women: Educational level was recorded as primary, secondary, vocational training or university studies. The occupational status was classified as inactive/unemployed, working or studying. Women working were asked for their occupational sector and their skill level. Occupations were classified according to the International Standard Classification of Occupations (ISCO)—08 as managers; professionals; technicians and associate professionals; clerical support workers; service and sales workers; skilled agricultural, forestry and fishery workers; plant and machine operators and assemblers; and elementary occupations.

Housing characteristics were self-reported, including surface in squared meters (later classified in four categories: <75/75–89/90–114/≥115), year of building (later categorized as before 1980/1980–1999/2000 or later), number of restrooms (classified as 1/2/≥3), number of rooms (classified as 1–2/3/≥4), number of people in the household (2/3/≥4), availability of heating and availability of air conditioning. Housing surface per person was classified by quartiles in ≤23 square meters/23.1–30.0/30.1–40.0/>40. Finally, we created a post hoc house score by adding a point for each of the following conditions: housing without heating, housing surface per person ≤23 square meters, number of rooms ≥4 and number of people in the household ≥4; thus, this score would rank from 0 to 4.

### 2.5. Detection of Women Positive to SARS-CoV-2

All women were tested for a SARS-CoV-2 current infection via PCR: women in sub-cohorts 1 and 2 at admission for delivery and women in sub-cohort 3 at recruitment. All women were tested for antibodies anti-SARS-CoV-2 via ELISA test (IgG, IgM and IgA) at recruitment. For the present analysis, women were considered positive to SARS-CoV-2 if they tested positive in either PCR or antibodies.

### 2.6. Statistical Analysis

Variables are described as number and percentage. The association between each socio-economic status variable and positivity to SARS-CoV-2 was assessed via logistic regression, always using the most frequent category of exposure as reference. The results are reported as odds ratios, whether crude or adjusted for age and country of origin. In order to analyze housing variables as continuous without assuming linear relationship with SARS-CoV-2 infection, we included them in the logistic regression models using cubic splines. Due to its post hoc nature, odds ratios and 95% confidence intervals for the housing score were obtained by bootstrapping with 1000 samples. All analyses were carried out with Stata 16/SE (Stata Corp).

## 3. Results

This analysis includes 988 women; 266 (28%) delivered their baby between 23 March and 25 May 2020 and were retrospectively recruited (sub-cohort 1). A total of 352 (37%) delivered from 26 May onwards and were prospectively recruited (sub-cohort 2). Finally, 332 women (35%) were recruited in their gestational 12th-week consultation after 26 May (sub-cohort 3). Table 1 displays their main characteristics. Only 50 women (5.1%) were younger than 25 and 103 (10.6%) were older than 40 years. Most women (848, 87.9%) were born in European countries. Only 642 women have already delivered, as most pregnancies in sub-cohort 3 are still ongoing. Among these 642 women, the neonate from this pregnancy was the first child for 348 women (54.2%), and about one out of three had previously had an abortion. Only 54 pregnancies (8.5) were obtained via assisted fertilization. Twenty-seven (4.3%) pregnancies ended before the 37th week, and 6 of them (1.0%) before the 34th week. Caesarean section occurred in 115 cases (18.1%) and other instrumental delivery in 42 (6.6%).

Table 2 shows a detailed description of variables related to socio-economic status. About half of the women had undertaken university studies (n = 450, 46.3%), and 148 (15.2%) only attainted primary studies. Regarding occupation status, 223 women (23%) were inactive or unemployed and 17 were students. Among the 733 women who were working, 247 (33.6%) were professionals, 164 of whom were health or teaching professionals, and 250 (34.1%) were service or sales workers, 100 of whom were personal care workers in health services. Most housings were built from the year 2000 onwards (n = 492, 57.1%), were bigger than 90 m^2^ (n = 534, 55.9%), had at least two restrooms (n = 632, 65.6%) and at least three rooms (n = 680, 69.6%), and were inhabited by four people or more (n = 352, 36.9%). Heating was present in 859 (88.2%) houses but air conditioning only in 30 (3.1%).

Thirty-seven women tested positive to SARS-CoV-2 (4.7%), 16 of whom were positive to RNA in the PCR test, 20 to IgG, 2 to IgM and 3 to IgA. The association between educational attainment, occupation and positivity to SARS-CoV-2 is shown in Table 3. No relevant differences were found regarding risk of positivity to SARS-CoV-2 across educational levels. When compared to service and sales workers, neither professionals altogether (OR = 0.84, 95% CI: 0.35–2.04) nor health or teaching professionals (OR = 0.79, 95% CI: 0.30–2.12) had higher risk of positivity. Personal care workers in health services showed higher risk (OR = 2.87, 95% CI: 0.84–9.79).

The results on the characteristics of housing are shown in Table 4 and Supplementary Appendix A. Housing surface (Table 4), year of building (Table 4) and number of restrooms (Table 4) were not associated with positivity to SARS-CoV-2, although the OR for surface smaller than 75 m^2^ was 2.00 (95% CI: 0.76–5.27), and the OR for year of building between 1980and 1999 was 2.09 (95% CI: 0.89–4.90). The number of rooms, however, was positively associated with higher risk of SARS-CoV-2 infection (OR = 2.07, 95% CI: 0.96–4.47 for housing with four or more rooms) (Table 4 and Appendix A). Four or more people in the household roughly doubled the risk of positivity to SARS-CoV-2 (OR = 1.91, 95% CI: 0.89–4.14). Lack of heating in housing increased the risk of SARS-CoV-2 positivity by 2.8 times (OR = 2.81, 95% CI: 1.24–6.34) (Table 4 and Appendix A). Houses with less than 23 square meters per person increased the risk of positivity to the virus by four times (OR = 3.97, 95% CI: 1.43–11.1) (Table 4 and Appendix A). Finally, the post hoc score combining a small house (≤23 square meters per person), a large amount of people in the household (≥4) and a large number of rooms (≥4) resulted in a strong association with positivity to SARS-CoV-2, with odds ratios 1, 1.8, 6.3 and 27.1 for scores 0 to 3 (*p* for trend < 0.001) (Table 4 and Appendix A).

## 4. Discussion

According to our results, positivity to SARS-CoV-2 in pregnancy is associated with some indicators of household quality, including a higher number of rooms, a higher number of people, lower surface area per person and a lack of heating. However, educational level, working/unemployed situation and type of work were not associated with positivity to coronavirus, with the exception of healthcare workers. Taking together the increase in risk associated with housing characteristics and the lack of association between working variables and coronavirus infection would support the hypothesis that pregnant women have been especially exposed to SARS-CoV-2 transmission in the household rather than in the workplace.

Spain has been one of the European countries most affected by the COVID-19 pandemic [9]. Partial lockdown was enforced from 16 March to 21 June, with total lockdown from 29 March to 11 April. A national study on the prevalence of antibodies anti-SARS-CoV-2 in the Spanish population did not find any difference according to the income level in its first wave, carried out in April-May 2020 [10] while Spain was under lockdown but reported a decreasing gradient in prevalence as income increased in the preliminary report of the fourth wave, which was carried out in November 2020 [11], with most economic sectors working. As far as family income is related with housing conditions, these results agree with ours, regarding higher risk of infection in people with worse housing conditions. Emeruwa et al. (2020) also found that the risk of SARS-CoV-2 infection increased in pregnant women living in neighborhoods that have a higher average of inhabitants per household and a higher ratio of inhabitants/rooms [7].

Contact with people with COVID-19 increases viral transmission, especially in the same household [12,13], where up to 59% contacts of an index case would have detectable SARS-CoV-2 at any time [14]. Housing conditions such as little space per person could have a deleterious effect on making self-isolation difficult and, thus, increasing time of interpersonal contact. In this regard, it is noteworthy that among different types of contacts—namely community, social, extended family and household contacts—those in the household are the only type showing an inter-generational pattern [13], which could have been responsible for larger transmission.

The only occupational group we found to be associated with higher risk of positivity to SARS-CoV-2 was healthcare workers. In this regard, most cases of transmission in the workplace seemed to be related to people working in residential care and healthcare workers [15,16]. It is noteworthy that lockdown, stay-at-home recommendations and non-essential work could have protected many workers from being infected with SARS-CoV-2 in the workplace, but it has not dealt with household transmission and could have even increased it [13].

Our results have some public health implications. Prevention measures in the workplace, including personal protective equipment or administrative controls, could have been generalized, and pregnant women could take maternity leave early in their pregnancy in order to avoid SARS-CoV-2 infection risk. In this regard, maternity leave in Spain is 100% retributed, and our data indicate that 40% women in sub-cohort 1 and 58% in sub-cohort 2 left before the 25th week of pregnancy (results not shown). Protection against SARS-CoV-2 transmission in the household is not that straightforward, and protective measures in the household, which could have been especially useful if women had taken maternity leave but their partner continued at work, have not been reinforced by public health authorities. Guidelines for quarantining household contacts with COVID-19 cases have been delivered [17,18], but enforcing them is difficult, as people could have limited resources or house space in order to accomplish them. Public health authorities could consider housing alternatives for pregnant women sharing a home with people with SARS-CoV-2.

Our study has some limitations. Firstly, our results combined both current and past SARS-CoV-2 infections, as we considered both positive RNA and antibody tests. In this respect, our results should be seen as cumulative in the whole studied period; therefore, we avoided any analyses by time period. Secondly, socio-economic information was self-reported, which might introduce some recall bias. As this information was recorded before testing for SARS-CoV-2 infection, we assume that such a bias—if it exists—should be non-differential, eventually leading to odds ratio estimations biased towards the null. Therefore, our findings on socio-economic factors associated with higher risk of SARS-CoV-2 infection can be considered as robust. Thirdly, some cells in Table 3 have a small number of subjects, leading to low statistical power. Therefore, negative results in some variables—namely, occupation—cannot be taken as proof of lack of association with SARS-CoV-2 infection. Finally, our house score was built post hoc; in spite of the internal validation obtained by bootstrapping, this score requires external validation in other cohorts. However, our study also has some strengths. Firstly, the recruited cohort is population-based, as our hospital has concentrated on almost all deliveries during the hottest months of the pandemic. Secondly, most samples were prospectively obtained. Finally, the sample size was close to 1000 pregnant women, with information obtained prospectively for about 80% of the participants.

Meteorological conditions could be associated with both SARS-CoV-2 circulation and a pregnant woman’s time at home, as well as her fulfilment of some recommendations such as regular ventilation. Further research is needed to ascertain whether the association we found between housing and risk of infection was similar in different waves of the pandemic, which occurred in different meteorological seasons.

## 5. Conclusions

In conclusion, the risk of infection by SARS-CoV-2 in Spanish pregnant women is associated with worse housing conditions but not with the occupational sector or educational attainment of the women. This suggests that measures against household transmission should be reinforced by pregnant women and their families. In this regard, guidelines on household quarantine could include specific recommendations if a pregnant woman lives in the household.

## Figures and Tables

**Table 1 ijerph-18-05133-t001:** Demographic and reproductive characteristics of women included in this analysis.

Variable	Category	N (%)
Age	<25 years	50 (5.1)
	25–34 years	482 (49.4)
	35–39 years	341 (34.9)
	≥40 years	103 (10.6)
Recruitment sub-cohort	Retrospective (delivery before 26 May)	266 (28.0)
	Prospective (delivery from 26 May onwards)	352 (37.0)
	Prospective (consultancy at 12th week of pregnancy from 26 May onwards)	332 (35.0)
Country of origin	European	848 (87.9)
	Central or Southern American	92 (9.4)
	Asian	8 (0.8)
	African	17 (1.8)
Number of children	1	348 (54.2)
	2	245 (38.2)
	≥3	49 (7.6)
Previous abortions	No	427 (66.5)
	Yes	215 (33.5)
Type of fecundation	Natural	581 (91.5)
	Assisted	54 (8.5)
Length of pregnancy	<34 weeks	6 (1.0)
	34–36 weeks	21 (3.3)
	≥37 weeks	603 (95.7)
Type of delivery	Eutocic	477 (75.2)
	Instrumental	42 (6.6)
	Caesarean section	115 (18.1)

**Table 2 ijerph-18-05133-t002:** Socio-economic characteristics of the women included in this analysis.

Variable	Category	N (%)
Educational level	Primary school	148 (15.2)
	Secondary school	65 (6.7)
	Vocational training	309 (31.8)
	University	450 (46.3)
Occupation	Unemployed/inactive	223 (23.0)
	Working	733 (75.5)
	Student	17 (1.5)
Occupation sector (only workers) n = 889 *	Managers	25 (3.4)
	Professional	247 (33.6)
	Of whom health or teaching professionals	164 (22.3)
	Technicians and associate professionals	24 (3.3)
	Clerical support workers	117 (15.9)
	Service and sales workers	250 (34.1)
	Of whom personal care workers in health services	100 (13.6)
	Skilled agricultural, forestry and fishery workers	7 (1.0)
	Plant and machine operators, and assemblers	14 (1.9)
	Elementary occupations	51 (6.9)
House: year of building	Before 1980	208 (24.2)
	1980–1999	161 (18.7)
	2000 or later	492 (57.1)
House: surface	<75 m^2^	227 (23.7)
	75–89 m^2^	196 (20.5)
	90–114 m^2^	279 (29.2)
	≥115 m^2^	255 (26.7)
House: number of restrooms	1	331 (34.4)
	2	500 (51.9)
	≥3	132 (13.7)
House: number of rooms	1 or 2	293 (30.4)
	3	487 (50.6)
	≥4	193 (19.0)
House: number of cohabiting	2	214 (22.4)
	3	388 (40.7)
	≥4	352 (36.9)
House: heating	No	115 (11.8)
	Yes	859 (88.2)
House: air conditioning	No	944 (96.9)
	Yes	30 (3.1)
Squared meters per person	≤23.0	250 (26.6)
	23.1–30.0	278 (29.6)
	30.1–40.0	199 (21.2)
	>40	213 (22.7)
Housing post hoc score	0	488 (49.9)
	1	399 (40.8)
	2	85 (8.7)
	3	6 (0.6)

* According to the International Standard Classification of Occupations (ISCO)—08. http://www.ilo.org/public/english/bureau/stat/isco/isco08/index.htm (accessed on 21 December 2020).

**Table 3 ijerph-18-05133-t003:** Association between educational/occupational characteristics and positivity to SARS-CoV-2 in pregnancy.

Variable	N Positive/N Negative	OR (95% CI)	OR (95% CI) *
Educational level			
Primary school	4/144	0.68 (0.23–2.03)	0.69 (0.22–2.11)
Secondary school	3/62	1.16 (0.33–4.06)	1.18 (0.31–4.46)
Vocational training	12/297	0.97 (0.46–2.04)	0.96 (0.45–2.03)
University	18/435	1 (ref.)	1 (ref.)
Occupation			
Unemployed/inactive	10/213	1.23 (0.58–2.58)	1.26 (0.58–2.75)
Working	27/706	1 (ref.)	1 (ref.)
Student	0/15	NA	NA
Occupation sector (only workers) n = 889			
Managers	0/25	NA	NA
Professional	10/237	0.92 (0.39–2.16)	0.84 (0.35–2.04)
Of whom health or teaching professionals	7/157	0.95 (0.37–2.47)	0–79 (0.30–2.12)
Technicians and associate professionals	1/23	0.92 (0.11–7.34)	0.84 (0.10–6.81)
Clerical support workers	5/112	0.96 (0.33–2.79)	0.90 (0.31–2.63)
Service and sales workers	10/242	1 (ref.)	1 (ref.)
Of whom personal care workers in health services	6/94	2.92 (0.86–9.93)	2.87 (0.84–9.79)
Skilled agricultural, forestry and fishery workers	0/7	NA	NA
Plant and machine operators, and assemblers	1/13	1.53 (0.19–12.5)	1.37 (0.17–11.4)
Elementary occupations	0/51	NA	NA

OR: odds ratio. CI: confidence interval. NA: not available. * Adjusted for age (continuous), country of origin (European/no European). ref.: reference.

**Table 4 ijerph-18-05133-t004:** Characteristics of the housing and positivity to SARS-CoV-2 in pregnancy.

Variable	N Positive/N Negative	OR (95% CI)	OR (95% CI) *
House: surface			
<75 m^2^	11/216	1.98 (0.75–5.19)	2.00 (0.76–5.27)
75–89 m^2^	7/189	1.43 (0.50–4.17)	1.46 (0.50–4.24)
90–114 m^2^	7/272	1 (ref.)	1 (ref.)
≥115 m^2^	11/244	1.75 (0.67–4.59)	1.72 (0.66–4.52)
House: Year of building			
Before 1980	4/204	0.62 (0.20–1.90)	0.73 (0.24–2.25)
1980–1999	9/152	1.88 (0.81–4.39)	2.09 (0.89–4.90)
2000 or later	15/477	1 (ref.)	1 (ref.)
House: number of restrooms			
1	15/316	1.53 (0.74–3.18)	1.60 (0.76–3.35)
2	15/485	1 (ref.)	1 (ref.)
≥3	6/126	1.54 (0.59–4.05)	1.51 (0.57–3.97)
House: number of rooms			
1 or 2	8/285	0.83 (0.35–1.96)	0.82 (0.34–1.94)
3	16/471	1 (ref.)	1 (ref.)
≥4	12/171	2.07 (0.96–4.46)	2.07 (0.96–4.47)
House: number of cohabiting			
2	7/207	1.16 (0.44–3.04)	1.11 (0.42–2.92)
3	11/377	1 (ref.)	1 (ref.)
≥4	18/334	1.85 (0.86–3.97)	1.91 (0.89–4.14)
House: heating			
No	9/106	2.52 (1.16–5.49)	2.81 (1.24–6.34)
Yes	28/831	1 (ref.)	1 (ref.)
House: air conditioning			
No	35/909	1 (ref.)	1 (ref.)
Yes	2/28	1.86 (0.42–8.10)	1.85 (0.42–8.06)
Squared meters per person			
≤23.0	16/234	3.73 (1.35–10.3)	3.97 (1.43–11.1)
23.1–30.0	5/273	1 (ref.)	1 (ref.)
30.1–40.0	7/192	1.99 (0.62–6.37)	1.96 (0.61–6.26)
>40	7/206	1.86 (0.58–5.93)	1.80 (0.56–5.76)
House score			
0	11/477	1 (ref.)	1 (ref.)
1	15/384	1.69 (0.77–3.73)	1.80 (0.81–4.00)
2	9/76	5.14 (2.06–12.8)	6.29 (2.45–16.2)
3	2/6	21.7 (3.59–131.1)	27.1 (4.23–173.1)

OR: odds ratio. CI: confidence interval. * Adjusted for age (continuous), country of origin (European/no European). ref.: reference.

## Data Availability

The data presented in this study are available on request from the corresponding author.

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
