# Peer review of "Influence of Socioeconomic Status on SARS-CoV-2 Infection in Spanish Pregnant Women. The MOACC-19 Cohort"

_ijerph, 2021, doi:10.3390/ijerph18105133_

Round 1

Reviewer 1 Report

The work addresses an emerging situation regarding the behavior of COVID 19 with professional practices and residential habitat in women. The main variables for problematizing these elements are exposed in the city of Santander, Spain. For the conclusions, it is debated whether the correlation between labour, and residential dimensions affect infection in pregnant women.

It is very important to highlight the average socioeconomic index of Santander, which must be within the highest in northern Spain. If housing characteristics are a factor, then the residential environment should also be considered.

Regarding the labour dimension, Table 3 is very clear in the dispersion of occupations, which helps to extrapolate levels of vulnerability around contagion. However, it remains to be resolved whether these activities transited into telematics modalities in the context of lockdown. Otherwise, the indicator is conceptually weak.

You should delve into the final considerations and projections of the study beyond the experiment at Santander. It is insisted that it should improve in discussion, where background.

To address with greater emphasis the analytical question of the results, particularly in the paragraph between Tables 3 and 4.

Reviewer 2 Report

As this article points that little is socioeconomic factors associated to SARS-CoV-2 infection in pregnant women. It is very important issue. 

However, the study has limitations.

Firstly, the persistence of Covid-19 may have a significant impact on the treatment and care of patients, so it is necessary to continue to strengthen research and remain vigilant in this regard.

Secondly, the severity of disease in patients with COVID-19 was recorded within 14 days of testing positive, but the likelihood of disease worsening after 14 days may not have been included in the analysis.

Thirdly, one of the main reasons COVID-19 is so contagious is that respiratory infections are easily spread. This means that in a good living and working environment with strong ventilation and large activity space, the infection rate can be reduced to a certain extent. It is obvious that this study concluded that housing characteris-tics, but not occupational or educational variables, were associated with SARS-CoV-2 infection.  It doesn't offer more details about socioeconomic factors associated to SARS-CoV-2 infection in pregnant women.  These details and characteristics only belong to pregnant women.

Based on the literature about COVID-19, there are some advice about how to prevent  COVID-19  in pregnant women? For example,  about housing prevention:

1, keep the room air fresh, the temperature is appropriate, timely open the window, avoid too cold or overheating, so as not to catch a cold.
2. Use towels, bath towels, tableware, bedding and other daily necessities for pregnant women separately to avoid cross-infection.
3. Keep hand hygiene at all times.
Wash hands with hand sanitizer or running soap before and after using the toilet, or use alcohol-based hand sanitizer;
Avoid touching your mouth, nose and eyes with your hands when in doubt about whether they are clean.
Cover your nose and mouth with a tissue when sneezing or coughing.
4. Maintain balanced nutrition, light diet, avoid excessive eating, and do a good job of weight control.
5. Avoid visiting relatives and friends, and avoid contact with people infected with respiratory tract and those who have been to areas with high epidemic incidence within two weeks.
6, maternal adhere to do a good job of breastfeeding, to wash hands correctly before breastfeeding.
7. Live a regular life, get enough sleep, drink plenty of water, exercise appropriately, keep a good attitude and enhance oneself resistance.

In addition, there are also more suggestion  about self-health monitoring  and matters needing attention when going out for medical treatment.

Reviewer 3 Report

It is a paper focused on the factors associated to SARS-CoV-2 infection in pregnant women. The introduction and method are adequate and the results are relevant, only the following is suggested:

  • Analyze the relevance of including figures 1, 2 and 3 since these variables were not positively associated to SARS-CoV-2.
  • Include proposals for future research in the discussion.
  • Review the list of references, so that they adhere to the format established in the APA manual. 

Reviewer 4 Report

This is a useful study which has been professionally conducted. In order to answer some of the questions for instance regarding the influence of level of education or professional category, however, a much larger sample would have been required. We can see in table 3 that the number of subjects in some of the cells has been really small. So some of the negative findings  are plagued by low statistical power. This should be pointed out in the discussion. It is always much more difficult to use absence of statistical significance for arguing lack of relationship than to use a significant finding for arguing that there is a positive relationship. Statistical power has to be taken into account.

In general the language flows well but a grammar check should be done since there are too many small errors her and there. I also discovered something which I think is an error. I am referring to the discussion where the following statement could be read:

Emeruwa et al (2020) also found that the risk of SARS-CoV-2 infection increased in pregnant women living in neighbourhoods that have higher average of inhabitants per household and lower ratio inhabitants / rooms[7]. 

I believe it should be ...and higher ratio inhabitants/rooms..

Round 2

Reviewer 2 Report

The revised manuscript has rewrited and made the modification and the supplement in the relevant part of the article. Although I still have a few doubts,  this article could be accepted after minor revision.  For example the Section Discussion is too long and must be simplified, focusing on topics that are closely related to the theme of the article.

Reviewer 4 Report

The ms has been appropriately improved

Author Response

RE: IJERPH-1198498. Influence of socioeconomic status on SARS-CoV-2 infection in Spanish pregnant women. The MOACC-19 cohort

Dear sir/madam,

Thank you very much for still considering our above-mentioned manuscript.